# The impact of COVID-19 in children with Sickle Cell Disease: Results of a multicentric registry

Claudia de Melo Oliveira[1]*, Victor Jablonski Soares[2], Josefina Aparecida Pellegrini Braga[3], Thaís Alcantara Bonilha[4], Isis Magalhães[5], Sandra Regina Loggetto[6], Ciliana Rechenmacher[1,7], Liane Esteves Daudt[1,7,8], Mariana Bohns Michalowski[1,7,9]

1 Departamento de Pediatria, Programa de Pós-Graduação em Saúde da Criança e do Adolescente, Universidade Federal do Rio Grande do Sul, Porto Alegre, Brazil, 2 Faculdade de Medicina, Universidade Federal do Rio Grande do Sul, Porto Alegre, Brazil, 3 Departamento de Pediatria, Universidade Federal de São Paulo, São Paulo, Brazil, 4 IPPMG Universidade Federal do Rio de Janeiro, Rio de Janeiro, Brazil, 5 Hospital da Criança de Brasília José Alencar, Brasília, Brazil, 6 Hospital Infantil Sabará, São Paulo, Brazil, 7 Laboratório de Pediatria Translacional, Serviço de Pesquisa Experimental, Hospital de Clínicas de Porto Alegre, Porto Alegre, Brazil, 8 Unidade de Hematologia e Transplante de Medula Óssea Pediátrica, Serviço de Hematologia Clínica, Hospital de Clínicas de Porto Alegre, Porto Alegre, Brazil, 9 Serviço de Oncologia Pediátrica, Hospital de Clínicas de Porto Alegre, Porto Alegre, Brazil

* claumeloo@gmail.com

## Abstract

### Objective

To analyze the outcomes of children with sickle cell disease (SCD) and COVID-19.

### Method

A multicenter prospective study was conducted in five hematological centers from Central and Southeast Brazil, starting in April 2020. The variables recorded include clinical symptoms, diagnostic methods, therapeutic measures, and treatment sites. The clinical repercussions of the infection on the initial treatment and the overall prognosis were also evaluated.

### Results

Twenty-five unvaccinated children, aged 4 to 17 years, with SCD and a positive SARS-CoV-2 RT-PCR result participated in this study. Patients were classified as SCD types SS (n = 20, 80%) and SC (n = 5, 20%). Clinical characteristics and evolution were similar in both groups (p>0.05), except for the fetal hemoglobin value which was higher among the SC patients (p = 0.025). The most frequent symptoms were hyperthermia (72%) and cough (40%). Three children were admitted to the intensive care unit, all of whom were overweight/obese (p = 0.078). No deaths were observed.

### Conclusions

Although SCD leads to specific complications, the results found in this sample suggest that COVID-19 does not seem to carry an increased mortality risk in pediatric patients with this disease.

**Data Availability Statement:** The data underlying the results presented in the study are available at https://doi.org/10.6084/m9.figshare.21453642.v1.

**Funding:** This work received financial supports from Conselho Nacional de Desenvolvimento Científico e Tecnológico (CNPq) and Coordenação de Aperfeiçoamento de Pessoal de Nível Superior (CAPES). This research was supported by the Research and Event Incentive Fund (FIPE) of the Porto Alegre Clínicas Hospital (HCPA).

**Competing interests:** The authors have declared that no competing interests exist.

## Introduction

Sickle Cell Disease (SCD) is considered the most common hemoglobinopathy worldwide and in Brazil [1]. It is estimated that 1,000 children are born with SCD in Brazil annually, thus, being the most prevalent genetic disease in the country [2]. SCD is caused by a mutation in the β-globin gene and results in the substitution of the amino acid valine for glutamic acid in the β-globin chain, leading to variant hemoglobin and thus dysfunctional erythrocytes. The homozygosity of the beta-S (β S) allele results in the clinical form "Sickle cell anemia", while the co-inheritance of HbS and HbC is known as HbSC. The disease manifests when Red blood cells (RBCs), which contain HbS or a combination of HbS with other variant β alleles, are exposed to a deoxygenated environment. Thus, these cells undergo polymerization and become rigid, becoming liable to haemolysis. The high density of these cells is responsible for altering the blood flow and vascular endothelial wall integrity, leading to disease manifestations such as acute chest syndrome (ACS) [3,4].

Patients with SCD have already been impacted by respiratory viruses that affected the human population in recent decades, such as H1N1, in the 2009 pandemic. During this period, one of the most frequent comorbidities in patients hospitalized was SCD because of their greater predispositions to organ complications and hypoxia [5]. Similarly, it has also been observed that adult patients (>19 years) with SCD may have a significantly higher risk of developing acute chest syndrome (ACS), a potentially fatal complication of COVID-19 [6,7].

Studies on COVID-19 have shown that symptomatic infections in children are rarer than in adults, and clinical manifestations in the former are usually milder [8]. Despite this, in a previous study conducted with children with cancer in Brazil, an increase in mortality among these patients was observed, in addition to an association between worse prognosis and nutritional status [9]. The present study aimed to evaluate the impact of COVID-19 on the pediatric population with SCD. Based on a prospective, national, and multicenter registry of children tested for SARS-CoV-2, this study described the clinical presentation and evolution of 25 children with SCD who were affected by COVID-19 and treated in reference centers in Brazil. Subsequently, their presentations were compared with other cases previously reported in the international literature. It was observed that neither the increase in the severity of infection nor the mortality in children with SCD nor the nutritional status of the patients affected the outcome. The relationship between obesity/overweight and the risk of admission to intensive care units (ICU) remains to be further studied.

## Methods

### Study population

From May 2020 to March 2021, five medical centers from Central, South, and Southeast Brazil collected clinical and laboratory data from pediatric patients (<18 years) with SCD who had tested positive for SARS-CoV-2 by real-time polymerase chain reaction (RT-PCR). Patients who were suspected of having COVID-19 due to clinical and/or radiological criteria but had negative results in molecular diagnosis were not included.

Data were recorded prospectively on the Redcap® software, thus ensuring confidentiality and allowing integration with other participating national institutions. The records were regularly assessed by the team responsible for the research for standardization purposes.

The variables included clinical symptoms, therapeutic measures, and treatment settings. Moreover, the impact the infection had on the baseline treatment and acute prognosis was evaluated.

The disease severity was classified according to the parameters established by Qiu et al [10]. Children who did not need oxygen therapy were classified as mild; those who needed

supplemental oxygen, as moderate or severe; and those who needed mechanical ventilation and/or were hemodynamically unstable, as critical. In this study, the patients admitted to the ICU were also considered critical. Nutritional status was determined according to the World Health Organization (WHO) classification, namely underweight (<-2DP), adequate weight, overweight (>+1 standard deviation [SD]–equivalent to body mass index [BMI] 25 kg/m2 at 19 years), and obesity (>+2 SD–equivalent to BMI 30 kg/m2 at 19 years) [11].

### Statistical analysis

Continuous variables were assessed using the Kolmogorov-Smirnov test of normality; those with non-normally distribution were expressed as medians and interquartile ranges. Qualitative variables were summarized as absolute and relative frequencies. The significance level adopted was 0.05. Data were compiled in a Microsoft Excel® spreadsheet and analyzed on the PASW Statistics® version 18.0 and WinPepi version 11.65 software. Fisher's exact test and Pearson's chi-square test were used for categorical variables, while Student's *t*-test and Mann-Whitney *U* test, for noncategorical variables. Log-rank tests (Mantel-Cox) were used to compare Kaplan-Meier survival curves between two or more groups.

This study was approved by the Research Ethics Committee of the Porto Alegre Clinical Hospital (HCPA—CAAE 33532320.4.1001.5327). The legal guardians of all participants provided written informed consent.

## Results

A total of 25 unvaccinated children with a mean age of 10.6 (4–17) years were included. Twenty (80%) presented hemoglobin SS disease (HbSS) and five (20%), SC. All participants were symptomatic at diagnosis. The most common symptoms included hyperthermia, cough, chest pain, and myalgia. Nine patients (36%) had oxygen saturation below 92% at diagnosis and four (16%) developed severe acute respiratory syndrome. According to the severity classification, eight patients were classified as mild and fourteen as moderate or severe. The three patients who needed monitoring in the ICU were classified as critical. The clinical and laboratory characteristics of the patients are summarized in Table 1. Two patients were treated in an outpatient regimen; twenty were admitted to an inpatient ward; and three, to an intensive care unit. Of the latter, one had a vaso-occlusive crisis, triggering an acute chest syndrome; another presented painful venous occlusions of the right lower limb and lumbar spine; and the third was monitored for presenting bone and abdominal pain. Fetal hemoglobin (HbF) levels were significantly higher in patients with Hb SS (p = 0.025).

Antibiotics were the most frequently prescribed drug: 44% of the patients received ceftriaxone; 24%, azithromycin; 16%, amoxicillin/clavulanate; 16%, ampicillin; and 20%, others. Corticosteroids (8%), oseltamivir (8%), and heparin (8%) were also used in some cases. No patients needed Remdesivir. Twelve patients were prescribed hydroxyurea (48%) as well.

Children were categorized according to their BMI into underweight, normal weight, overweight, and obesity. These data were analyzed taking into account ICU admissions. The three patients who required admission to the ICU (p = 0,078) were among the nine children who were overweight or obese.

## Discussion

To date, there is little data in the literature on the presentation of COVID-19 in children with SCD. In the present study, the clinical behavior of the infection in this population was described, and a favorable evolution with no related deaths was observed. Thus far, this is one of the largest studies in Latin America to observe the evolution of children with SCD and

Table 1. Characteristics of patients.

| Characteristics | Total n = 25 | SC n = 5 | SS n = 20 | p value |
|---|---|---|---|---|
| **Sex** | male = 12 <br> female = 13 | male = 4 <br> female = 1 | male = 8 <br> female = 20 | 0.311 |
| **Age years (q25-q75)** | 10.56 (8–13) | 9.6 (8–11) | 10.8 (9–14) | - |
| **BMI** <br> **Under weight** <br> **Normal weight** <br> **Overweight** <br> **Obese** | <br> 2 <br> 14 <br> 5 <br> 4 | <br> 0 <br> 2 <br> 0 <br> 3 | <br> 2 <br> 12 <br> 5 <br> 1 | 0.053 |
| **Symptoms related to COVID 19 (%)** <br> Hyperthermia <br> Cough <br> **Chest Pain** <br> **Oxygen Saturation <92%** <br> **Severe Acute Chest** <br> **Respiratory Dysfunction** <br> Headache <br> Myalgia | <br> 18 (72%) <br> 10 (40%) <br> 9 (36%) <br> 9 (36%) <br> 4 (16%) <br> 3 (12%) <br> 3 (12%) <br> 2 (8%) | <br> 3 (60%) <br> 0 <br> 1 (20%) <br> 1 (20%) <br> 1 (20%) <br> 1 (20%) <br> 1 (20%) <br> 0 | <br> 15 (75%) <br> 10 (50%) <br> 8 (40%) <br> 8 (40%) <br> 3 (15%) <br> 2 (10%) <br> 2 (10%) <br> 2 (10%) | <br> 0.548 <br> **0.040*** <br> 0.360 <br> 0.360 <br> 1.000 <br> 0.411 <br> 0.411 <br> 1.000 |
| **Leukocytes** <br> **Hemoglobin** <br> **Platelets** <br> **PNN** <br> **Lymphocytes** <br> **PCR** <br> **D Dimers** <br> **LDH** <br> **Hb F** | 17055.82 (n = 22) <br> 8.461 (n = 23) <br> 304028.3 (n = 22) <br> 11719.8 (n = 21) <br> 3854.67 (n = 21) <br> 115.2 (n = 17) <br> 2566.9 (n = 14) <br> 1230.13 (n = 8) <br> 10.7 (n = 24) | 12746.7(n = 3) <br> 9.05 (n = 4) <br> 223666.7 (n = 3) <br> 12268.5 (n = 2) <br> 2686.7 (n = 3) <br> 210.7 (n = 3) <br> 5.99 (n = 1) <br> 1314(n = 2) <br> 5.45 (n = 4) | 17736.2(n = 19) <br> 8.34 (n = 19) <br> 316717 (n = 19) <br> 11662.0 (n = 19) <br> 4049.3 (n = 18) <br> 94.73 (n = 14) <br> 2763.9 (n = 13) <br> 801.3(n = 13) <br> 15.05 (n = 20) | 0.427 <br> 0.302 <br> 0.368 <br> 0.894 <br> 1.00 <br> 0.149 <br> 0.885 <br> 0.144 <br> **0.025*** |
| **ICU** | 12.0%(n = 3) | 40%(n = 2) | 5.0%(n = 1) | 0.117 |
| **Oxygen therapy** <br> **Severity** <br> Mild <br> Moderate/ severe <br> Critical | 59.1%(n = 13/22) <br> 32% (n = 8/25) <br> 56% (n = 14/25) <br> 12% (n = 3/25) | 60%(n = 3/5) <br> 40%(n = 2/5) <br> 20%(n = 1/5) <br> 40%(n = 2/5) | 58.8%(n = 10/17) <br> 30% (n = 6/20) <br> 65% (n = 14/20) <br> 5% (n = 1/20) | 1.0 |

BMI—Body Mass Index; ICU—Intensive Care Unit; Hb–Hemoglobin.

COVID-19. Previously, Panepinto et al. [7] studied a group of 178 people, of which 48 were children, from the SECURE-SCD Registry between March and May 2020. Only one child died. The authors also analyzed adults and reported that approximately 40% of deaths occurred among patients who had milder SCD-associated types (HbSC or HbSβ + thalassemia types) and a history of very frequent pain episodes, pulmonary hypertension, decreased renal function, sickle cell neuropathy, and stroke in the previous three years. Although in our study there was no statistical difference between the groups, it is noteworthy that two patients out of five with HbSC required ICU admission, while only one out of 20 patients with HbSS needed it. Studies that include more children should be carried out so that a definitive conclusion can be reached in this aspect.

In general, the literature indicates that children with SCD and other hemoglobin disorders have fewer symptoms compared to adults [12]. A study conducted in France investigated patients with COVID-19 and SCD and concluded that those over 45 years of age were more likely to have more complications and be admitted to the ICU [6]. Our data presented comparable results as the children evaluated had few complications associated with the infection and only three were admitted to the ICU. This study is somewhat limited given that, for instance,

the questionnaires were completed by local teams at each facility rather than centrally. For this reason, careful monitoring of the data was required to ensure uniformity and consistency. Despite our best efforts, there was a considerable lack of data for some patients. This is also due to the fact that patients with mild symptoms stayed at home without any further medical care, so data were not collected during this period.

It is interesting to notice that, at the beginning of this trial, a much higher number of patients with SCD was expected to require hospitalization for SARS-CoV-2 infection. This assumption was based on the previous experience during the influenza H1N1 pandemic [13]. In a study with 40 children with SCD who were diagnosed with H1N1, 50% of them were admitted to the hospital and 25% developed acute chest syndrome. The hospitalization rate was higher than that of the general population, which was 7%. Moreover, SCD was one of the most frequent comorbidities among patients hospitalized then [5,14]. A retrospective cohort conducted at Johns Hopkins Hospital analyzed cases of influenza A, B, and H1N1 in children and young adults with SCD and found that these patients needed intensive care more often and received more blood transfusions and antiviral treatments, in addition to developing acute chest syndrome and having episodes of intense pain [15]. Studies have shown that the predis-position to organ complications and hypoxia is greater in patients with SCD affected by respi-ratory viruses [5,13–15]. The pathophysiology of SCD could play the key role in this fact. SCD manifests due to vaso-occlusion, endothelial dysfunction, inflammation and infections. The chronic hemolysis and altered morphology of erythrocytes increases the plasma viscosity, con-tributing to impaired blood flow and promoting vaso-occlusion. Also, both endothelial dys-function and inflammation upregulate molecules of selectins (P- and E-), vascular-cell-adhesion molecule-1 (VCAM-1), ICAM-1, and major leukocyte chemoattractants such as KC (in mice) or interleukin-8 (IL-8) (in humans) on endothelial cells, while the activation of leu-kocytes leads to increased adhesion of these cells to the activated endothelium. Since COVID-19 induces pulmonary, cardiac, and peripheral thrombotic events, it was expected that both diseases would lead to severe manifestations and, thus, hospitalizations [3].

In a previous study, this group of researchers analyzed 179 children with cancer and SARS-CoV-2. Nutritional status also impacted the prognosis [9]. In the present case series, the three patients who required treatment in the ICU were classified as overweight and/or obese. A sta-tistically significant difference might not have been observed due to the small number of par-ticipants; however, there seems to be a trend toward a worse prognosis among overweight/obese patients.

In conclusion, in our sample of children with SCD, COVID-19 did not lead to an increase in mortality rates. This result appears to be different from those previously found for other respiratory viral infections known to cause serious complications in these patients.

## Acknowledgments

The authors would like to thank the contributions of the investigators from all participating centers across the country, as well as the Brazilian Association of Hematology, Hemotherapy and Cell Therapy (ABHH) for supporting the promotion of this study.

## Author Contributions

**Data curation:** Claudia de Melo Oliveira, Victor Jablonski Soares, Josefina Aparecida Pelle-grini Braga, Thaís Alcantara Bonilha, Isis Magalhães, Sandra Regina Loggetto, Ciliana Rechenmacher, Liane Esteves Daudt, Mariana Bohns Michalowski.

**Supervision:** Liane Esteves Daudt, Mariana Bohns Michalowski.

**Writing – original draft:** Claudia de Melo Oliveira, Victor Jablonski Soares, Josefina Apare-
cida Pellegrini Braga, Sandra Regina Loggetto, Ciliana Rechenmacher.

**Writing – review & editing:** Liane Esteves Daudt, Mariana Bohns Michalowski.

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
