## [Decision Letter · Decision Letter 0]

14 Aug 2022

PONE-D-22-15661The impact of COVID-19 in children with Sickle Cell Anemia: results of a multicentric registry.PLOS ONE

Dear Dr. Oliveira,

Thank you for submitting your manuscript to PLOS ONE. After careful consideration, we feel that it has merit but does not fully meet PLOS ONE’s publication criteria as it currently stands. Therefore, we invite you to submit a revised version of the manuscript that addresses the points raised during the review process.

Please check the reviewers' remarks.

We look forward to receiving your revised manuscript.

Kind regards,

Marcus Tolentino Silva

Academic Editor

PLOS ONE

Journal Requirements:

Reviewers' comments:

Reviewer's Responses to Questions

**Comments to the Author**

1. Is the manuscript technically sound, and do the data support the conclusions?

Reviewer #1: Yes

Reviewer #2: Partly

2. Has the statistical analysis been performed appropriately and rigorously? 

Reviewer #1: Yes

Reviewer #2: Yes

3. Have the authors made all data underlying the findings in their manuscript fully available?

Reviewer #1: Yes

Reviewer #2: Yes

4. Is the manuscript presented in an intelligible fashion and written in standard English?

Reviewer #1: Yes

Reviewer #2: No

5. Review Comments to the Author

Reviewer #1: The authors present the short-term outcomes of pediatric patients with Sickle Cell Anemia who had symptoms of Covid-19 prior to vaccination. Data were obtained from more than one treatment center in Brazil, located in the Midwest, Southeast and South regions. The results were properly analyzed and discussed, with objectivity.

Suggestion to authors: remove lines 2-9, page 8, from the Introduction, as they contain information that is already well known and unnecessary considering the profile of the journal's readers. The paragraph between lines 13-19 is important for contextualization and could be developed. In the same way, line 66, page 10, Results, must be deleted as it is already found in Method.

Having made these considerations, the manuscript is ready for publication.

Reviewer #2: The main objective of the manuscript is to describe the behavior and evolution of sickle cell disease patients who had real-time PCR test positive for SARS-CoV2. The study included patients from Central and Southeastern regions of Brazil, late the authors include the South, and it needs to be included in the abstract, and presents data for one year since the diagnosis of COVID-19 was realized.

General concept comments

The manuscript covers a field that still have little knowledge of, that is the COVID-19; and when it comes to COVID-19 in sickle cell disease, this scarcity of data is even greater, given the complexity of the two diseases. In this way, the focus of the work is important and pertinent. What I consider an area of weakness in the manuscript is really the number of individuals included in the study, 25 in total, 20 with sickle cell anemia and 5 with SC disease, a very small number for the conclusions presented by the authors. However, this reviewer will make the considerations to assist in the general conception of the analysis of the manuscript. In addition, this reviewer believe that it is important to discuss a little more about the pathophysiology and repercussions of sickle cell disease, both in the introduction and in the discussion, especially in the latter, since the most clinical results found were not discussed, which in my view do not match the conclusion placed by the authors that “In conclusion, COVID-19 in children with SCA did not lead to an increase in mortality rates, unlike other respiratory viral infections known for causing serious complications in these patients”, since the number of patients followed up does not allow the authors to reach such a conclusion.

Firstly, there is a mistake in the use of the term sickle cell disease, as the series comprises individuals with sickle cell anemia (HbSS) and SC disease (HbSC), the authors should change the general term to sickle cell disease, including both groups investigated. This change must also occur in the title and keywords, which must be changed to sickle cell disease.

The authors describe the symptoms presented by patients who have a very diverse clinical picture, some classified as moderate to severe. The authors also refer to data loss and data entry, which may have caused some bias in the study. Considering that the number of patients analyzed is extremely small, it will be important to be very careful with the conclusions. Another important aspect is related to the previous data of these patients; did they have many symptoms before COVID? What about laboratory tests? What were the units of each variable presented on table 1? Were the laboratory methods similar in all institutions participating in the study?

Since the study took a year to collect data, do you hear a follow-up of patients? Today we know that long-term and post-COVID-19 are reality, how did these patients maintain themselves during the study period? It is these questions that demonstrate that the study, despite being well structured, still has questions to be presented, providing a clear overview of who these patients are and how they evolved with the development of the disease. Another aspect is the comparison between laboratory data between SS and SC individuals with COVID, is SC expected to be less severe? This can be an interesting question, but it needs a more robust number of individuals to be well answered.

Were these patients previously treated with hydroxyurea? Could this make a difference in the response to SARS-CoV-2 infection?

Please explain the study design, what it means “a multicenter observational cohort study was conducted on prospective data and retrospective analyses”, it was not clear.

Regarding the table presented in the manuscript, the table 1, it will be important that some numbers be revised, as they are not currently presented in English.

6. PLOS authors have the option to publish the peer review history of their article (what does this mean?). If published, this will include your full peer review and any attached files.

Reviewer #1: **Yes: **Rosana Cipolotti

Reviewer #2: No

---

## [Author Response · Author response to Decision Letter 0]

1 Nov 2022

First of all we would like to thank the reviewers for the time dedicated and the opportunity of answering these questions and improving our paper. They were very relevant to qualify the information we wanted to show. I approached each of their concerns very carefully.

Please find here below our answers and some supplementary data.

Changes requested by reviewers that should be added to the article were entered in the file and are highlighted in yellow as well as the answers to the reviewers' questions are attached in a single file.

Best regards,

Claudia

---

## [Decision Letter · Decision Letter 1]

6 Dec 2022

PONE-D-22-15661R1The impact of COVID-19 in children with Sickle Cell Disease: results of a multicentric registry.PLOS ONE

Dear Dr. Oliveira,

Thank you for submitting your manuscript to PLOS ONE. After careful consideration, we feel that it has merit but does not fully meet PLOS ONE’s publication criteria as it currently stands. Therefore, we invite you to submit a revised version of the manuscript that addresses the points raised during the review process.

ACADEMIC EDITOR: please check all minor reviews pointed out by the reviewer. 

We look forward to receiving your revised manuscript.

Kind regards,

Marcus Tolentino Silva

Academic Editor

PLOS ONE

Journal Requirements:

Reviewers' comments:

Reviewer's Responses to Questions

**Comments to the Author**

1. If the authors have adequately addressed your comments raised in a previous round of review and you feel that this manuscript is now acceptable for publication, you may indicate that here to bypass the “Comments to the Author” section, enter your conflict of interest statement in the “Confidential to Editor” section, and submit your "Accept" recommendation.

Reviewer #1: All comments have been addressed

Reviewer #2: (No Response)

2. Is the manuscript technically sound, and do the data support the conclusions?

Reviewer #1: Yes

Reviewer #2: Yes

3. Has the statistical analysis been performed appropriately and rigorously? 

Reviewer #1: Yes

Reviewer #2: Yes

4. Have the authors made all data underlying the findings in their manuscript fully available?

Reviewer #1: Yes

Reviewer #2: Yes

5. Is the manuscript presented in an intelligible fashion and written in standard English?

Reviewer #1: Yes

Reviewer #2: No

6. Review Comments to the Author

Reviewer #1: The authors have adequately addressed all the comments raised in the previous round of review and this manuscript is acceptable for publication

Reviewer #2: The authors developed a careful review of the manuscript and emphasized aspects that this reviewer considered relevant. The authors initially included a first paragraph about sickle cell disease, which complements the manuscript and supports the study that was developed.

As stated in the analysis carried out earlier, this reviewer considers that the data presented provide important information about covid-19 in sickle cell disease, reinforcing the need for additional studies aimed at learning more about this interaction. However, this reviewer still felt the need for some corrections in the text, as well as a grammar review so that the manuscript is uniform in terms of the choice of English language.

Regarding to the manuscript, I noticed that there is a need for some adjustments throughout the text.

1- It will be important that in the first citation of sickle cell disease, the author already includes the acronym SCD and already includes it in the text from that moment.

2- In the first paragraph the authors still in the first paragraph of the introduction section, in line 11 in the text "It is estimated that 1,000 children are born with it in Brazil annually", please replace it with "children are born with SCD", it is important to mention it in the sentence and not leave as a hidden subject.

3- line 14, first paragraph of the introduction, replace abnormal hemoglobin with variant hemoglobin.

4- in general, the text mixes American English with British English, authors should choose one of the two and keep the language constant in the text.

5- In the session called statistical analysis, should be used for description of normally and non-normally distributed data, and not asymmetric distribution.

6- Lines 72-73, use the same description mentioned before, the authors do not need to repeat, use HbSS and HbSC.

7- in the text include in line 123, the authors mention that studies have shown about the predisposition of organs to hypoxia, but only include a single reference, it will be important to mention others or adjust the sentence for that single reference.

7. PLOS authors have the option to publish the peer review history of their article (what does this mean?). If published, this will include your full peer review and any attached files.

Reviewer #1: **Yes: **Rosana Cipolotti

Reviewer #2: No

---

## [Author Response · Author response to Decision Letter 1]

20 Jan 2023

Dear Emily Chenette

First of all we would like to thank the reviewers for the time dedicated and the opportunity of answering these questions and improving our paper. They were very relevant to qualify the information we wanted to show. I approached each of their concerns very carefully.

Changes requested by reviewers that should be added to the article were entered in the file and are highlighted in yellow in the file "manuscript with track changes".

Best Regards,

Claudia

---

## [Decision Letter · Decision Letter 2]

15 Feb 2023

The impact of COVID-19 in children with Sickle Cell Disease: results of a multicentric registry.

PONE-D-22-15661R2

Dear Dr. Oliveira,

We’re pleased to inform you that your manuscript has been judged scientifically suitable for publication and will be formally accepted for publication once it meets all outstanding technical requirements.

Kind regards,

Marcus Tolentino Silva

Academic Editor

PLOS ONE

Additional Editor Comments (optional):

Reviewers' comments:

Reviewer's Responses to Questions

**Comments to the Author**

1. If the authors have adequately addressed your comments raised in a previous round of review and you feel that this manuscript is now acceptable for publication, you may indicate that here to bypass the “Comments to the Author” section, enter your conflict of interest statement in the “Confidential to Editor” section, and submit your "Accept" recommendation.

Reviewer #1: All comments have been addressed

Reviewer #2: All comments have been addressed

2. Is the manuscript technically sound, and do the data support the conclusions?

Reviewer #1: Yes

Reviewer #2: Yes

3. Has the statistical analysis been performed appropriately and rigorously? 

Reviewer #1: Yes

Reviewer #2: Yes

4. Have the authors made all data underlying the findings in their manuscript fully available?

Reviewer #1: Yes

Reviewer #2: Yes

5. Is the manuscript presented in an intelligible fashion and written in standard English?

Reviewer #1: Yes

Reviewer #2: Yes

6. Review Comments to the Author

Reviewer #1: No more comments to the Authors because all the previous comments have been addressed. The Authors achieved their research goals.

Reviewer #2: The authors answers all reviewer questions. The review believe the manuscript is now read to be published in Plos One, if the Editors agree with this.

This review just ask the authors to change on the first paragraph, please write hemolysis in the American way.

7. PLOS authors have the option to publish the peer review history of their article (what does this mean?). If published, this will include your full peer review and any attached files.

Reviewer #1: **Yes: **Rosana Cipolotti

Reviewer #2: No

---

## [Editor Report · Acceptance letter]

28 Feb 2023

PONE-D-22-15661R2 

The impact of COVID-19 in children with Sickle Cell Disease: results of a multicentric registry 

Dear Dr. Oliveira:

I'm pleased to inform you that your manuscript has been deemed suitable for publication in PLOS ONE. Congratulations! Your manuscript is now with our production department. 

Kind regards, 

on behalf of

Dr. Marcus Tolentino Silva 

Academic Editor

PLOS ONE